# SALIENT EXPLANATION FOR FINE-GRAINED CLASSIFICATION

## ABSTRACT

Explaining the prediction of deep models has gained increasing attention to increase its applicability, even spreading it to life-affecting decisions. However there has been no attempt to pinpoint only the most discriminative features contributing specifically to separating different classes in a fine-grained classification task. This paper introduces a novel notion of salient explanation and proposes a simple yet effective salient explanation method called Gaussian light and shadow (GLAS), which estimates the spatial impact of deep models by the feature perturbation inspired by light and shadow in nature. GLAS provides a useful coarse-to-fine control benefiting from scalability of Gaussian mask. We also devised the ability to identify multiple instances through recursive GLAS. We prove the effectiveness of GLAS for fine-grained classification using the fine-grained classification dataset. To show the general applicability, we also illustrate that GLAS has state-of-the-art performance at high speed (about 0.5 sec per $224 \times 224$ image) via the ImageNet Large Scale Visual Recognition Challenge.

## 1 INTRODUCTION

Over the last several years, convolutional neural networks (CNNs) (LuCun et al., 2015) have achieved superior performance in various computer vision tasks, including image classification (He et al., 2016; Shi et al., 2018), object detection (Oh et al., 2017; Zhou et al., 2016), semantic segmentation (Pathak et al., 2015), and image captioning (Xu et al., 2015). Despite these dramatic advances, the opacity of CNNs makes it difficult to understand why they reach particular decisions, limiting the ability to widen their application to various fields.

In general, the visual interpretation of deep learning models is understood as estimating the impact of a particular neuron activation related to a given input instance. In white-box approach, architectural modification of the classification model (Bach et al., 2015; Dong et al., 2017; Mahendran & Vedaldi, 2016; Selvaraju et al., 2017; Simonyan et al., 2013; Springenberg et al., 2014; Zhou et al., 2016; Zeiler & Fergus, 2013) or access to specific layers (Bach et al., 2015; Selvaraju et al., 2017; Zhang et al., 2016) is inevitable (Petsiuk et al., 2018), resulting in severe limitation of application. In contrast, the black-box approach (Seo et al., 2018; Petsiuk et al., 2018; Ribeiro et al., 2016; Tian & Cai, 2017; Zeiler & Fergus, 2013; Zintgraf et al., 2017; Fong & Vedaldi, 2017) aims to be inherently model agnostic. Its main concerns are how to perturb an input image and draw the model's response on the perturbed instance to the final heat map. For example, the Randomized Input Sampling for Explanation (RISE) method (Petsiuk et al., 2018) perturbed an image with a randomised mask to measure the importance of pixels and then linearly fused all importance from several thousand masks.

The conventional black-box methods employed unnatural and fragile perturbation schemes such as single colour out (Seo et al., 2018; Petsiuk et al., 2018; Ribeiro et al., 2016; Zeiler & Fergus, 2013), random noise (Tian & Cai, 2017; Zintgraf et al., 2017; Fong & Vedaldi, 2017) and smoothing (Fong & Vedaldi, 2017). These perturbation schemes have several limitations. First, they are deficient in pinpointing only the most discriminative, i.e., salient features that are essential for the fine-grained classification tasks where the between-class shape similarity is very high; for example, pinpointing only the red face of Red-faced Cormorant in the bird classification task in Figure 1 is crucial for explaining why a deep learning model classifies the image as Red-faced Cormorant. Second, the conventional perturbation schemes highly suffer from local noise and, thus, fuse maps from a con-

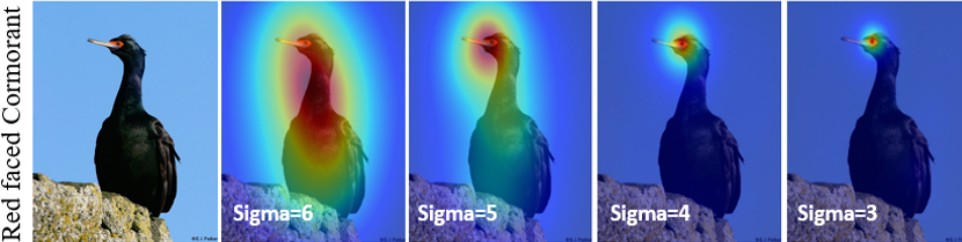

Figure 1: Salient explanation. As the value of  decreases, the heat map concentrates more and more on the most discriminative part of the relevant objects, the bird's face in this case. The salient explanation is very important for a variety of tasks such as the fine-grained classification and biomarker discovery in medical image. Note that the salient explanation is possible due to our idea of adopting Gaussian mask.

siderable number of perturbations for a reliable explanation. This is the main cause of slowness with conventional black-box methods.

Inspired by the lighting and shadowing phenomena in nature, we propose a simple yet effective black-box method, called Gaussian light and shadow (GLAS), which simulates feature perturbation as the presence or the absence of light at the pixel level of an image. The primary idea of GLAS is to perturb an input image by the Gaussian mask (light) and inverse-Gaussian mask (shadow) and, then, record the responses of the perturbed images. GLAS uses a simple grid search; once completed over the entire image, the response maps are fused to construct the final heat map. The fusion mimics the Gaussian mixture. The proposed method has several advantages compared with other black-box methods (Petsiuk et al., 2018; Zintgraf et al., 2017; Fong & Vedaldi, 2017). First, GLAS provides scalability of explanation that we can achieve by adjusting the variance parameter of the Gaussian mask. The scalability makes it possible to pinpoint clues for the salient explanation, which is not feasible with the nonparameterized approaches (Mahendran & Vedaldi, 2016; Zhou et al., 2016; Zeiler & Fergus, 2013; Petsiuk et al., 2018; Zintgraf et al., 2017; Fong & Vedaldi, 2017). The salient explanation is valid for explaining fine-grained classifications, such as classifying bird species in a CUB200 dataset, involving large between-class similarity and significant within-class variance. Figure 1 shows an image of the Red-faced cormorant species that we can discriminate by identifying the face color. It illustrates that GLAS adjusts its gaze from the body to the red face as the scale parameter decreases and finally pinpoints the red area around the eye. Second, our pixel-wise multiplication operation with the Gaussian mask at a specific search point simulates the gradual dimming effect as going farther from the center. We argue that because of this characteristic, a significantly reduced number of perturbations is sufficient. GLAS can process an image much faster than conventional methods.

To summarize, the contributions of this paper are as follows: We introduce a novel notion of salient explanation which is critical in explaining the fine-grained classification tasks. We propose a simple yet efficient black-box method, GLAS, which provides an easy way to perturb an input image based on Gaussian lighting and shadowing. (1) GLAS is fast because of the smoothly varying shape of the Gaussian mask, which generates a visual explanation up to one order of magnitude faster than other black-box methods. (3) We show the broad applicability of GLAS to various other tasks: object localization and visual captioning. Quantitative comparisons show that GRAS is superior to conventional methods.

## 2 RELATED WORKS

The white-box approach heavily uses the network's internal information, such as gradients or feature maps of specific layers. A gradient can indicate how much a small change in a pixel influences the class output (Springenberg et al., 2014). For example, Simonyan et al. (2013) proposed the gradient-based model, which directly mapped saliency values to the original space. Additionally, Zeiler & Fergus (2013) proposed a deconvolution method. In the method, the forward signal is reversed at a neuron and backpropagated to the input space. The study (Bach et al., 2015) proposed the layer-wise relevance propagation method, in which the prediction in the output layer is decomposed into pixel-wise relevance values and backpropagated until it satisfies the conservation rule. Samek et al. (2016)

emphasized the importance of quantitative evaluation and provided a rigorous comparison of the previously mentioned methods; these approaches are extensively reviewed in Samek et al. (2016). The visual feature maps provide important clues for the explanation, which some techniques exploit. The technique (Zhou et al., 2016) called class activation mapping (CAM) is accomplished by weighted fusion of visual feature maps and requires the modification of CNN architecture, replacing the fully connected layer with the global average pooling. Grad-CAM (Selvaraju et al., 2017), an extended version of CAM, is applicable to a broader range of CNNs. The previously mentioned techniques modify the model's internal operations or rely on the model's internal values; thus, they are model dependent.

The black-box approach measures the response change of the base model when the input instance is spatially perturbed, and this change can be regarded as the significance of the classifier's decision. The study (Robnik & Kononenko, 2008) simulates feature perturbation based on marginal probability, and several studies have extended and improved this method (Tian & Cai, 2017; Zintgraf et al., 2017). For the CNN-based architectures, the study (Zintgraf et al., 2017) proposed the conditional sampling approach. The method considers that a given pixel value highly depends on neighboring pixels and that multivariate analysis excludes a rectangular region rather than a single pixel. Because a pixel-wise perturbation method such as random noise (Zintgraf et al., 2017; Fong & Vedaldi, 2017) was considered, pixels are highly vulnerable to adversarial attack.

Several techniques (Seo et al., 2018; Ribeiro et al., 2016; Tian & Cai, 2017) aim at region-wise perturbation approaches, rather than using pixels. A study (Tian & Cai, 2017) improved the conditional sampling method using the superpixel algorithm, making it more robust from local noise than Zintgraf's method. Additionally, the superpixel segmentation technique was used in existing methods (Seo et al., 2018; Ribeiro et al., 2016; Tian & Cai, 2017). In these methods, high-level segments, rather than pixels (Zintgraf et al., 2017) or oversegmented regions (Tian & Cai, 2017), are used to perturb the feature of instance, and the methods have achieved explanation results that are more visually pleasing compared with previous methods. However, the results are probably limited when the segmentation map's quality is poor. Petisiuk et al. proposed the RISE method (Petsiuk et al., 2018), which simulates the feature's absence using randomized masks and measures its response to each masked instance. Because of its random masking strategy, RISE requires a considerable number of feedforward executions and suffers from local noise. The meta learning approach that tries to maximize the interpretability of a learning model is used in some studies (Ribeiro et al., 2016; Fong & Vedaldi, 2017). One study (Ribeiro et al., 2016) employed superpixel-wise random samples around the instance and an approximate linear decision model. Fong & Vedaldi (2017) proposed an optimized framework that learns a minimum perturbation mask from the corresponding response to its output neuron. However, such frameworks often fail to optimize their result because of its sensitivity to various types of models and instances. Unlike the white-box approaches, the black-box methods are inherently model agnostic; i.e., they are applicable to any learning model, because they rely only on the output values, regardless of the internal workings of the classification models.

The fine-grained classification is to recognize subordinate classes of a base class such as species of birds and different models of cars and planes. Most of recent works use the deep CNN models and propose better loss function. Shi et al. (2018) proposed a generalized large-margin (GLM) loss to reduce between-class similarity and within-class variance. The contrastive loss (Sun et al., 2014) and triplet loss (Schroff et al., 2015) have also been proposed. Qiu et al. 2018 proposed a method based the sqeeze-and-excitation atention model. Peng et al. 2018 used both the object-level attention and part-level attention. The literatures treated various types of objects. Shi et al. (2018) used birds, cars, and airplanes datasets. Other objects include fashion (Seo et al., 2018), fish (Qiu et al., 2018), vehicle (Li et al., 2019), plant (Lin et al., 2019), leukemia (Sipes & Li, 2018), and plankton (Lee et al., 2016).

## 3 PROPOSED APPROACH

**GLAS method.** Given an image $\mathbf{I}$, we define a set of search points $\Omega = (\mu_1, \mu_2, ..., \mu_{k \times k})$ by the centers of $k \times k$ grids overlaid upon $\mathbf{I}$, as shown in Figure 2. For a given class label $y$ and a specific search point $\mu_i$, the prediction score $f_l(\mu_i)$ by Gaussian light can be written as

$$f_l(\mu_i) = P(y|\mathbf{I} \odot G(\mu_i, \sigma_l)) \tag{1}$$

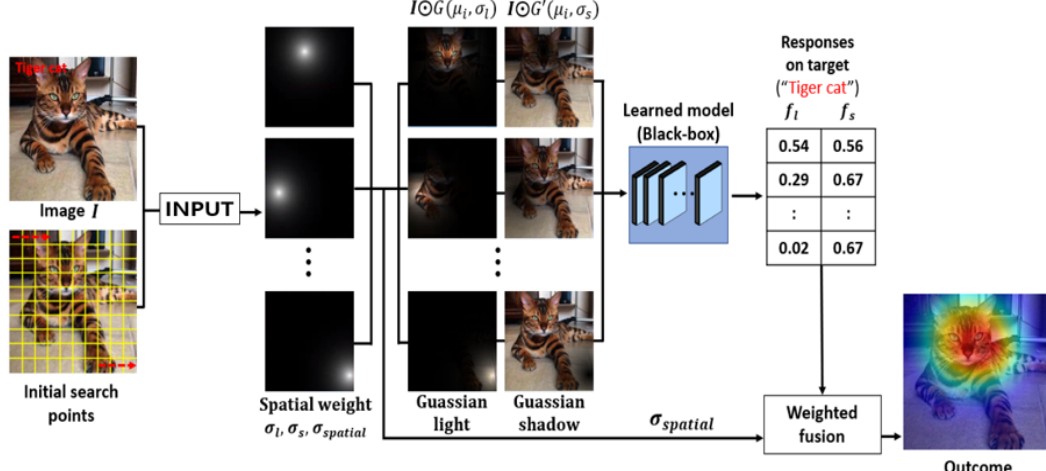

Figure 2: Overview of the GLAS method.

Where $\odot$ denotes element-wise multiplication, and $G(\mu_i, \sigma_l)$ is a Gaussian distribution with mean $\mu_i$ and standard deviation $\sigma_l$. Eq 1 simulates light projected on a specific part of the image to measure the contribution of the local pattern. It is also possible to define the score based on the inverse-Gaussian mask; i.e., the shadow is given by the following equation:

$$f_s(\mu_i) = |P(y|\mathbf{I}) - P(y|\mathbf{I} \odot G'(\mu_i, \sigma_l))| + \lambda \qquad (2)$$

where $G'(\mu_i, \sigma_l) = 1 - G(\mu_i, \sigma_l)$. Here, $\lambda$ is a constant, and we empirically set it to $10^5$ to avoid $f_s(\mu_i)$ being 0. We use a weighted fusion to define the saliency score $S(x_j)$ for a pixel $x_j$ as

$$S(x_j) = \frac{1}{|\Omega|} \sum_{\mu_i \in \Omega} exp(-\frac{D(x_j, \mu_i)}{\sigma_{spatial}^2}) f_l(\mu_i) f_s(\mu_i) \qquad (3)$$

where $exp(-\frac{D(x_j,\mu_i)}{\sigma_{spatial}^2})$ is a spatial weighting factor; here, $D(a, b)$ denotes the distance between $a$ and $b$. Eq3 represents the Gaussian mixture-based weighted fusion. The high flexibility of the visual explanation can be achieved by adjusting the scale parameter for each Gaussian mask.

**Recursive GLAS (RGLAS) method.** GLAS tends to high-light the most discriminative clue. To discover the various evidences that lead to the classifier's decision, we propose a simple schema called RGLAS. Figure 3 shows the key idea of RGLAS: to prevent revisits to the search points related to the most discriminative fea-tures that have already been found, leading to extraction of the next important features. This mechanism also helps in discovering multiple instances in an

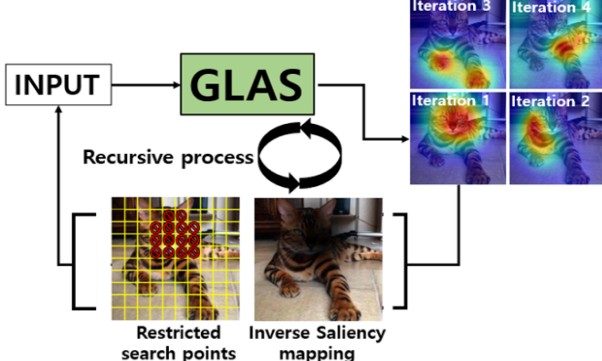

Figure 3: Framework of RGLAS. The GLAS instances are repeated until all discriminative patterns related to a given class have been dis-covered.

---

**Algorithm 1** :Recursive GLAS

---

**Require:** image $I$, target class $c$
**Ensure:** Final saliency map **O**
  1:  **O**=0; $\Omega = (\mu_1, \mu_2, ..., \mu_{k \times k})$
  2:  **while** $\sqrt{mean(S)/f(y|I)} <$t$_2$ **do**
  3:     **for** each pixel $x_j$ **do**
  4:       $S(x_j) = \frac{1}{|\Omega|} \sum_{\mu_i \in \Omega} exp(-\frac{D(x_j, \mu_i)}{\sigma^2_{spatial}}) f_l(\mu_i) f_s(\mu_i)$
  5:     **end for**
  6:     $B = S > t_2$
  7:     **for** each serach point $\mu_i$ **do**
  8:       If($B(\mu_i)$==1) remove $\mu_i$ from $\Omega$
  9:       $I = I \odot (1 - S)$
10:     **end for**
11:     **O**$+ = normalizeS$

---

image. The RGLAS algorithm starts by constructing the saliency map $S$ (lines 3–4). We compute the binary map of S using the threshold value $t_1 = 0.8$ (line 5) and eliminate search points located in the positive region of the binary map (lines 6–7). The input image is updated using the previous input and the inverse saliency map (line 8). We define a simple stop condition, as formu-lated in line 9, with $t_2 = 5$. We found that as the iteration increases, $mean(S)$ tends to increase but $f(y|I)$ decreases, guaranteeing that the stop condition occurs consistently.

## 4 EXPERIMENTAL RESULTS

The experiments were conducted on an Intel Core i7-7800X with a 3.50 CPU, 32 GB of memory, and a GTX 1080 Ti GPU. We aimed to evaluate quantitatively and qualitatively the salient explanation capabilities of GLAS and existing explanation models.

**Salient explanation for fine-grained classification tasks.**

The GLAS provides us with fine-level visual clue identification, enabling the salient explanation. To demonstrate the effectiveness of scalability, we employed CUB200 (Wah et al., 2011), Stanford Cars (Krause et al., 2013), and Air-craft (Maji et al., 2013) benchmarks that have been used for the fine-grained image classification tasks. The CUB200 dataset consists of 11,788 images of 200 bird species. The Stanford Cars dataset includes 8,144 training and 8,041 test images with 196 classes. The Aircraft dataset is a set of 10,000 images with 100 classes reflecting a fine-grained

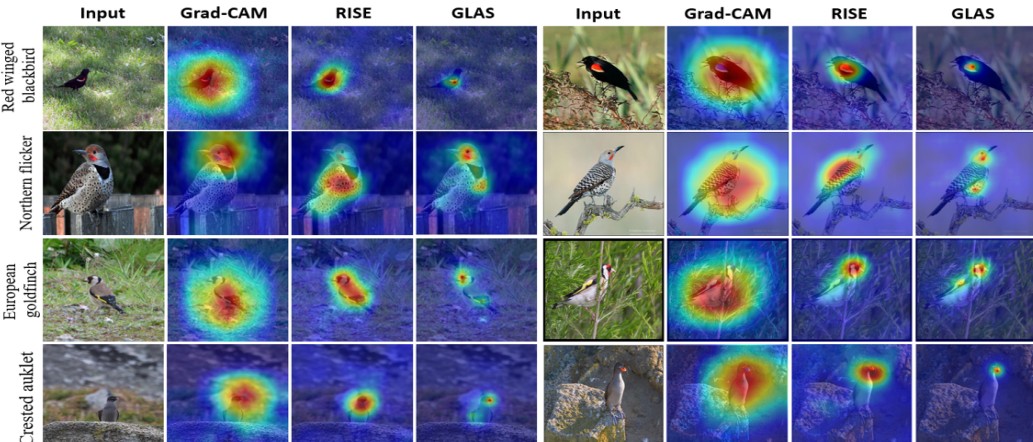

Figure 4: Visual comparison with the existing models. From top to bottom: red-winged blackbird, Northern flicker, European goldfinch, and crested auklet. We introduce the uniform characteristics of each bird: first row (red wings), second row (red below the eyes and black spots on the body), third row (red face and yellow on the wings), and fourth row (orange beak).

set of airplanes. We used the basic procedure of transfer learning using ResNet-50 pretrained on ILSVRC. The resulting networks of CUB200, Stanford Cars and Aircraft yielded top-1 accuracies of 76.11%, 92.01%, and 80.59%, respectively. GLAS can pinpoint more detailed clues by adjusting its scale parameters.Figure 4 shows the visual comparison. In the experiment, we used an equal sigma value of 3.0 for $\sigma_l, \sigma_s$, and $\sigma_{spatial}$. Unlike other methods, we can see that GLAS consistently pinpoints meaningful patterns of birds. For example, in the case of the Northern flicker, an instance has characteristics such as the red spot below the eyes and black dots on the body. GLAS surprisingly pinpoints two characteristics of the Northern flicker with the scale 3.0; however, the other methods only discover the location of the instance and fail to explain the detailed patterns. In this regard, Grad-CAM and RISE tend to explain only global significance, such as the target's location. Numerous visual examples are available in the Supplementary Materials.

We conducted another experiment measuring how much the pinpointed clues affect the class decision. In Figure 5, original images are perturbed by the inverse heat map, and the amount that the score drops is recorded. As expected, the class score dropped rapidly, ensuring the significance of the pinpointed features. Figure 6 shows the European goldfinch characterized by red face and yellow spot on the wings used for demonstrating coarse-to-fine controls. GLAS adjusts the standard deviation. Deconv adjusts the occlusion mask size. The RISE adjusts the size of the initial mask. The most prominent observation is that the heat maps from RISE and Deconv are very noisy and less accurate in identifying the most discriminative parts of the relevant object. The failure case analysis in Figure 7 deserves an attention. The second row unveils an interesting behavior of CNN through failure cases. The first and second images belonging to the Red-legged Kittiwake was incorrectly classified into Pigeon guillemot with 34.09% and 11.59% probabilities, respectively. The salient explanation capability of GLAS allows us to understand that the CNN misclassified the images into the Pigeon guillemot by looking at the red leg. Figure 8 shows the visual explanations on Aircraft and Stanford Cars benchmarks, respectively. For Aircraft examples, GLAS consistently pinpointed propellers of wings for the class "Yu 12". For Stanford Cars dataset, we found that CNN changed its gaze consistently according to poses of Car. When the front is shown for the car class 138, the grille part is highly probable to be pinpointed. When the back side is shown, lamps and wheels are pinpointed. Note that these behaviors of CNNs can be unveiled only when the salient explanation is available.



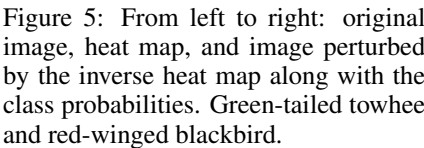

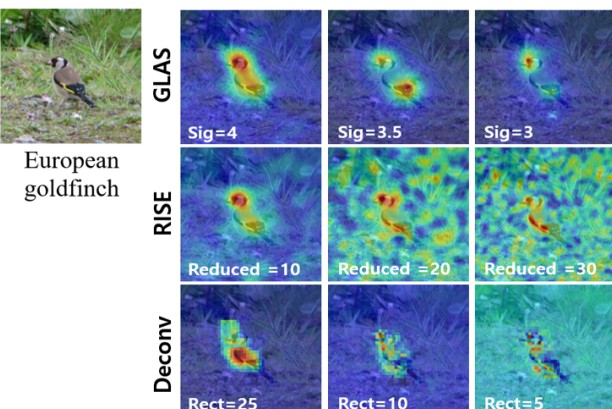

Figure 5: From left to right: original image, heat map, and image perturbed by the inverse heat map along with the class probabilities. Green-tailed towhee and red-winged blackbird.

Figure 6: Visual comparisons of GLAS, RISE, and Deconv according to their parameters controlling the locality. The example is the Eu-ropean goldfinch characterized by red face and yellow spot on the wings. GLAS adjusts the standard deviation. Deconv adjusts the occlusion mask size. The RISE adjusts the size of the initial mask.

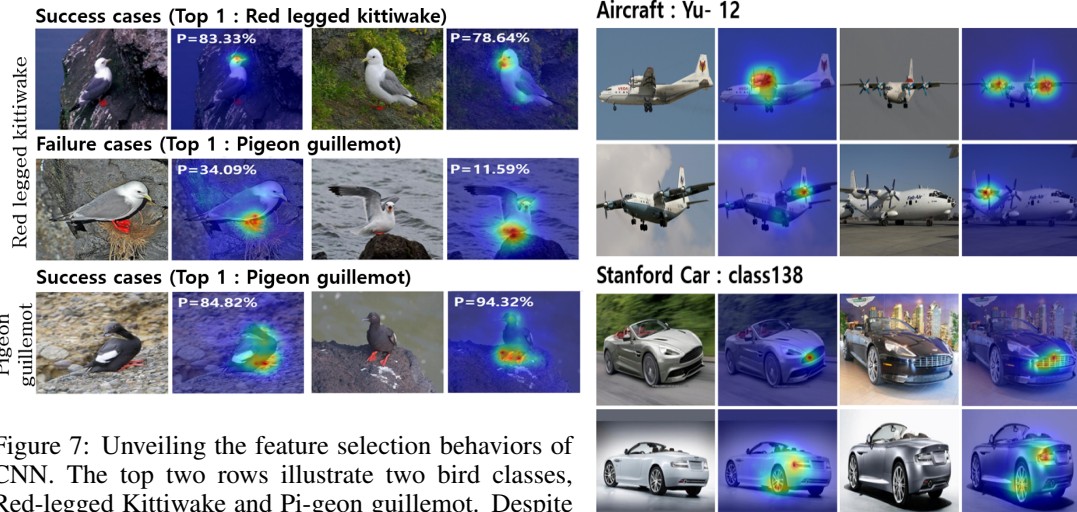

Figure 7: Unveiling the feature selection behaviors of CNN. The top two rows illustrate two bird classes, Red-legged Kittiwake and Pi-geon guillemot. Despite its name, the Red-legged Kittiwake were consistently highlighted on faces while the Pigeon guil-lemot was highlighted on the red legs

Figure 8: Visual explanations on Aircraft and Stanford Car benchmarks.

**Evaluation on target localization.**

We performed quantitative evaluations on the ImageNet Large Scale Visual Recognition Challenge (ILSVRC) (Deng et al., 2009). As a metric, we employed the pointing game (PT) presented in the study (Ribeiro et al., 2016) . The PT purely measures the spatial selectiveness of the continuous visual saliency map. In the evaluation, the PT detects the maximum intensity point on the saliency map, and a Hit is recorded if the maximum point is in the ground-truth annotation; otherwise, a Miss is recorded. The accuracy is calculated using $Acc = \frac{Hit}{Hit+Miss}$. Because multiple maximum points often arise, we em-ployed the threshold value T > 0.95 to generate binary blobs, and then we used the centroid of the biggest blob as the localization point. We empirically set the scale parameters $\sigma_l$=5 and $\sigma_s$=3, with $\sigma_{spatial}$=6.

Table 1: PT scores according to the number of search points

| Search points $k \times k$ | $12 \times 12$ | $15 \times 15$ | $22 \times 22$ | $25 \times 25$ | $30 \times 30$ |
|---|---|---|---|---|---|
| PT score | 0.905 | 0.912 | 0.914 | 0.913 | 0.913 |
| Time(s) | 0.398 | 0.652 | 1.435 | 1.909 | 3.95 |

Table 2: Quantitative comparisons with existing models using the PT on the ILSVRC validation data using ResNet50

| Methodn | PT | PT-small | Time(s) |
|---|---|---|---|
| Grads (Simonyan et al. 2013) | 0.773 | 0.604 | 0.128 |
| Deconv (Zeiler et al. 2014) | 0.750 | 0.584 | 0.117 |
| Grad-CAM (Selvaraju et al. 2017) | 0.901 | 0.754 | 0.183 |
| Deconv (Zeiler et al. 2014) | 0.809 | 0.688 | 2.64 |
| LIME (Ribeiro et al. 2016) | 0.766 | 0.645 | 15.11 |
| RISE (Petisiuk et al. 2018) | 0.907 | 0.787 | 8.11 |
| MASK (Fong et al. 2017) | 0.841 | 0.711 | 16.38 |
| GLAS (light) | 0.889 | 0.782 | 0.328 |
| GLAS (shadow) | 0.877 | 0.751 | 0.328 |
| GLAS (fusion) | 0.912 | 0.792 | 0.612 |

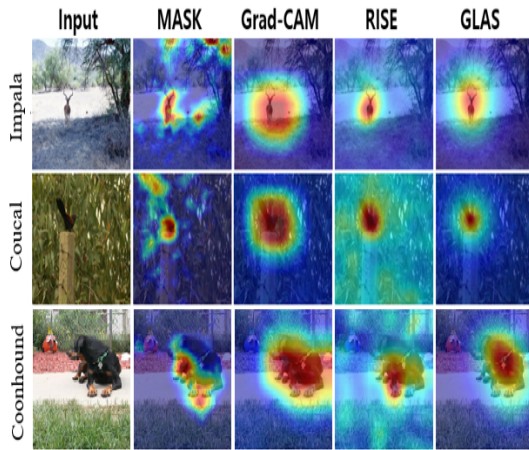

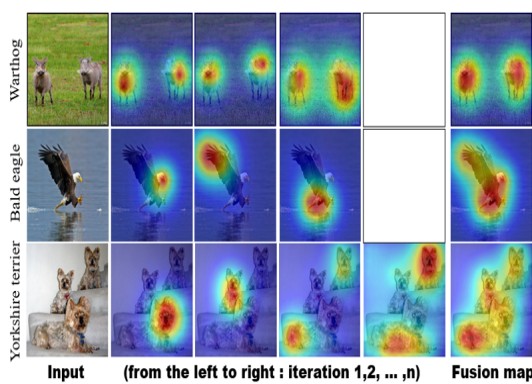

Figure 9: Visual comparison of the class-discriminative capability of MASK, Grad-CAM, RISE, and GLAS by varying the object classes.

Figure 10: Recursive process. This algorithm discovers the relevant parts of a given class in order of significance. We used the adaptive stop condition mentioned in Section 3.

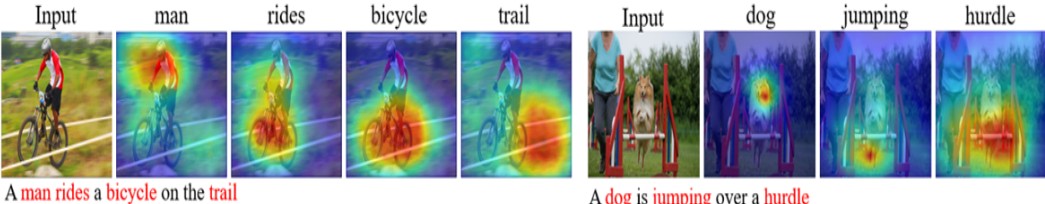

Figure 11: Visual explanation examples produced by GLAS for the image captioning model.

Table 1 shows the execution time of GLAS according to the number of grid search points. The result tends to show favorable PT scores as the number of the search points grows. It starts to become saturated after $k = 15$ in terms of performance, even as the execution time continues to grow. Table 2 illustrates the results of comparisons with the existing methods on the ILSVRC validation dataset. GLAS achieves the best result in terms of PT score. GLAS is 13 times faster than RISE even with the higher PT score. This is because a considerable number of perturbed images using a randomized masking process are necessary for reliable visual explanation in RISE. When RISE is forced to use 450 ($225 \times 2$) perturbed images, identical to the number used by GLAS, we observed that the PT score of RISE drops from 0.907 to 0.869. This observation tells us that GLAS perturbs and localizes the important features efficiently. We separately evaluated the cases in which the object is small in the PT-small column of 2. Like a related study (Cao et al., 2015), we consider an object to be small if the total area of the bounding box of the given class is smaller than one quarter of the size of the image. Even though all models encountered a performance drop, GLAS still beats other models.

In our work, GLAS operations can be used together or independently. The results in the last three rows of Table 2 show an ablation study on GLAS by measuring model performance with either Gaussian lighting or shadowing suppressed. Table 2 shows the performance of the ablated GLAS is comparable to that the state-of-the-art methods, outperforming all methods except Grad-CAM, RISE, and fused GLAS. Figure 9 provides a visual comparison of the methods. GLAS and Grad-CAM clearly highlight the important region related to a given class, whereas MASK and RISE suffer from nontrivial local noise. The advantage of GLAS is obvious without the noise and produces a visualization map that is highly interpretable. Because the GLAS map consists of Gaussian mixture clues, it identifies the most important area without being distracted by meaningless clues.

**Multiple evidence discovery by the recursive process.**

Table 3 illustrates the mean intersection over union (IOU) scores of the proposed recursive process. Because the visual saliency map consists of continuous intensity values, the mean IOU scores were measured with varying thresholds, from 0 to 1.0. In particular, significant progress occurs at the second iteration. Higher IOU scores over the threshold values indicate that the object regions are

Table 3: IOU scores of the proposed method on the ILSVRC validation data as the iteration increases. "A" represents the final iteration under the adaptive stop condition

| Iteration | Threshold | | | | |
|---|---|---|---|---|---|
| | 0.1 | 0.3 | 0.5 | 0.7 | 0.9 |
| 1 | 0.53 | 0.51 | 0.40 | 0.31 | 0.09 |
| 2 | 0.54 | 0.56 | 0.51 | 0.39 | 0.07 |
| 3 | 0.55 | 0.56 | 0.52 | 0.37 | 0.06 |
| A | 0.55 | 0.59 | 0.55 | 0.38 | 0.06 |

uniformly highlighted. In the fusion results shown in the last column of Figure 10, we can see that multiple instances and multiple evidences are well discovered.

**Visualization of the image captioning model.**

Image captioning is a challenging task for which both computer vision and natural language processing techniques should be considered. We constructed the image captioning model based on publicly available implementations for which the fine-tuned InceptionV3-based image and long short-term memory-based language models are considered. Figure 11 shows some visual explanation results from the image captioning model to demonstrate the applicability of GLAS. GLAS shows the capability to localize visual concepts such as objects (man, bicycle, ball, boy, hurdle, and dog) and actions (riding, playing, and jumping).

## 5 CONCLUSION

In this study, we proposed a visual explanation method called GLAS for the black-box model. Our method is in-spired by the natural light and shadow phenomena and provides a simple yet robust way to perturb an input in-stance. The GLAS presented the ability of fine-level visual explanation at various scales through the adjustment of the Gaussian scale. Additionally, we showed the broad applicability of GLAS to various tasks. In experiments, GLAS showed state-of-the-art performance and efficient computing time. For a future work, we plan to improve the heat map by optimizing the scale parameters of the Gauss-ian mask adaptively on an image instance. A deeper theo-retical analysis of GLAS is also needed.

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

## 6 APPENDIX

**A. Visualization of neuroimaging classification model.**

We consider a neuroimaging classification problem to show applicability of GLAS in a medical imaging domain. We employed 3D-magnetic resonance imaging (MRI) scans reflecting 199 Alzheimer's Disease (AD) vs. 230 healthy Normal Control (NC) from the Alzheimer's Disease Neuroimaging Initiative (ADNI), which is publicly available. A 3D-CNN was employed for classification of AD vs. NC. Due to limited dataset size, 3D-MRIs are spatially normalized based on template brain image and unsupervised learning technique (Convolutional auto-encoder) is applied before supervised learning. The overall architecture is comprised of three 333 Conv layers with 10 filters each, two FC layers with 32 and 16 nodes each, and softmax activation; each of Conv layers is followed by ReLU and 222 max-pooling; Gaussian dropout with a dropping ratio 0.8 is applied in between Conv layers; in the FC layers, we used SELU activation for speeding up learning and taking normalization effect in the FC layers. 5-fold cross validations were conducted to evaluate the classification model. The mean accuracy was 85.31%.

Since the Automated Anatomical Labeling (AAL) [5] map exists, all MRIs are spatially normalized based on AAL, and then we considered the centroid of AAL as the search points. In this work, 3D-GLAS method were considered to perturb 3D-MRI data, and we empirically set $\sigma$=10 and $\sigma$=15. The validation MRIs of AD category are fed to 3D-GLAS, then the entire saliency maps of validation MRIs were linearly integrated and normalized. For this reason, the highlighted biomarkers have statistical significance. Since the centroids of each AAL segment are considered as search points, we mark out impact of each biomarker to its corresponding AAL segment directly. In Figure 25, the hippocampus, amygdala and temporal inf were selected as the important biomarkers for the accurate classification of AD. These biomarkers have previously been known to be closely related with dementia in many existing studies [1][2][3][4]. In particular, the hippocampus, a brain region for learning and memory, is one of the first brain biomarker, which is affected by AD and undergoes severe structural changes as the disease progresses [4]

[1] R. Casanova, C.T. Whitlow, B. Wagner, J. Williamson, S.A. Shumaker,J.A. Maldjian M.A. Espeland. High Dimensional Classification of Structural MRI Alzheimer's Disease Data Based on Large Scale Regularization. Frontiers in Neuroimformatics, doi:0.3389/fninf.2011.00022. 2011.

[2] H. Kilian, T. Vinh-Thong, C. Gwenaelle, T. Thomas, V.M. Jose C. Pierrick. Multimodal Hippocampal Subfield Grading for Alzheimer's Disease Classification. doi: https://doi.org/10.1101/293126. 2018.

[3] S.W. Seo, N. Ayakta, L.T. Grinberg, S.Villeneuve, M. Lehmann, B. Reed G.D. Rabinovici. Regional correlations between [11C] PIB PET and post-mortem burden of amyloid-beta pathology in a diverse neuropathological cohort. Neuroimage Clinic 13 (Suppl. C), 130–137,2017.

[4] S. Tian, C Ying. Visualizing Deep Networks with Interaction of Super-pixel with Interaction of Super-pixels. In CIKM, 2017.

[5] N. Tzourio-Mazoyer, B. Landeau, D. Papathanassiou, F. Crivello, O. Etard, N. Delcroix, Bernard Mazoyer M. Joliot. Automated anatomical labeling of activations in SPM using a macroscopic anatomical parcellation of the MNI MRI single-subject brain. NeuroImage 15: 273-289, 2002

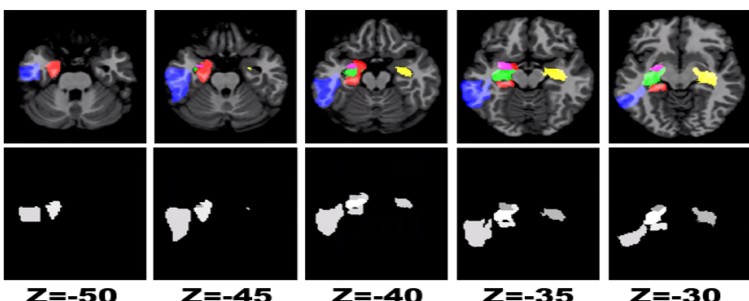

Figure 12: Visual distribution of discriminative biomarkers in the classification of AD. The first rows illustrate a brain template image overlapped with important biomarkers (rank 1 of Table 1: red, rank 2: green, rank 3: blue, rank 4: yellow, rank 5: purple). The second rows show AAL wise saliency map.

**B. Salient explanation results for fine-grained classification task.**

In this part, we provide numerous visual examples to show the precision explanation. We chose 13 bird categories of CUB200 that have the unique characteristics. In the experiment, we used the same sigma value of 3.0.

**Red winged blackbird**

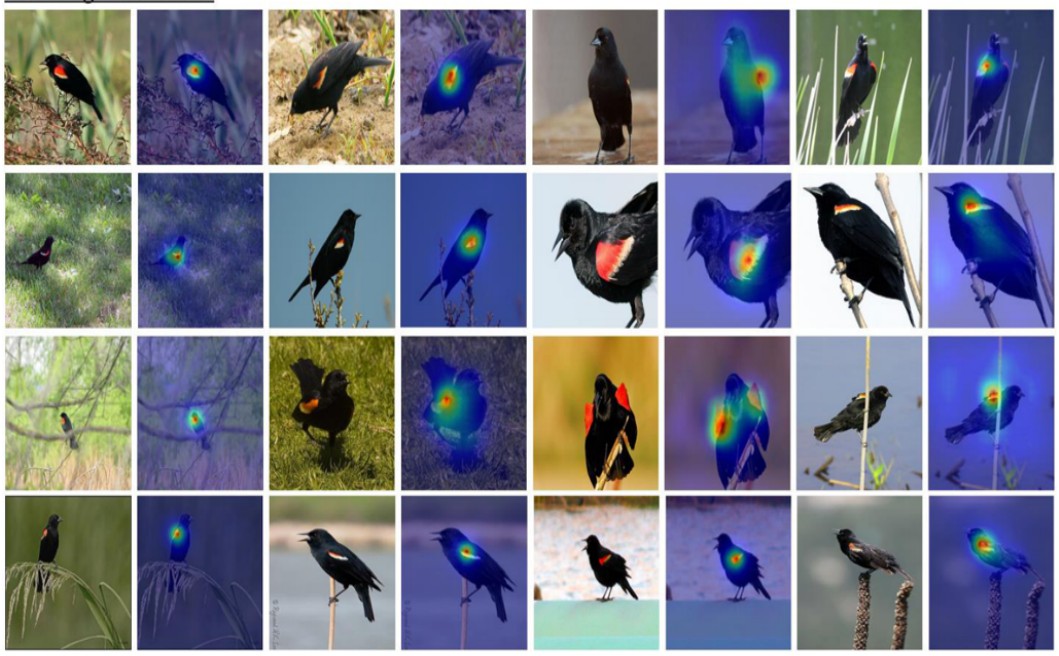

Figure 13: Red winged blackbird. The uniform characteristic is red shoulder patch.

**Northern flicker**

Figure 14: Red winged blackbird. The uniform characteristic is red shoulder patch.

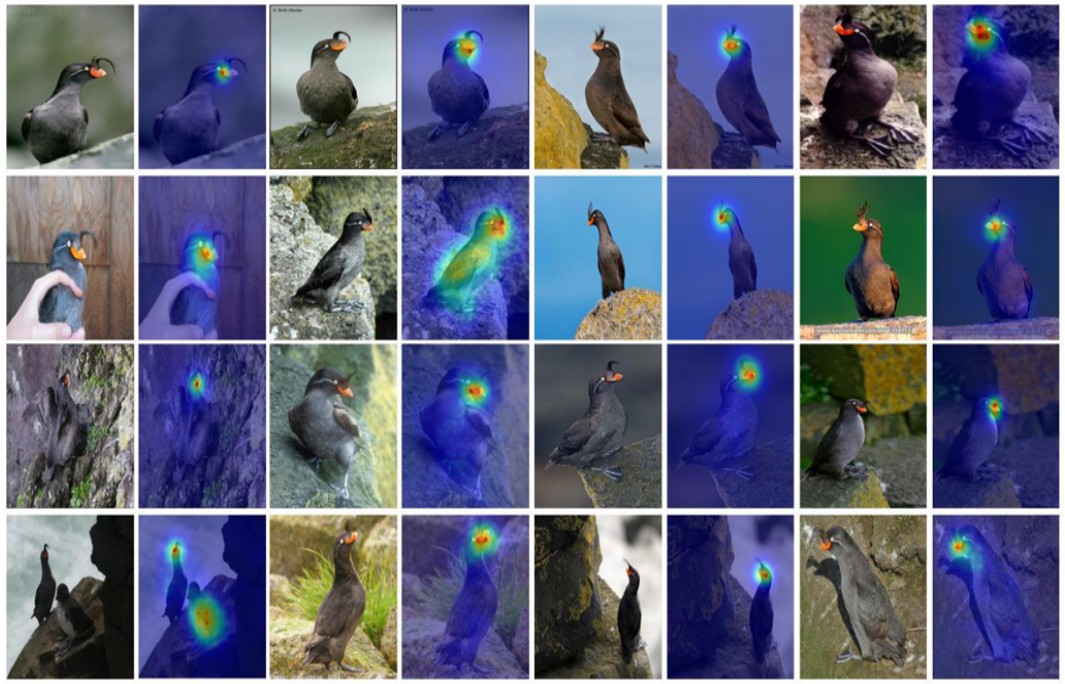

Figure 15: Crested auklet. The uniform characteristic is orange beak.

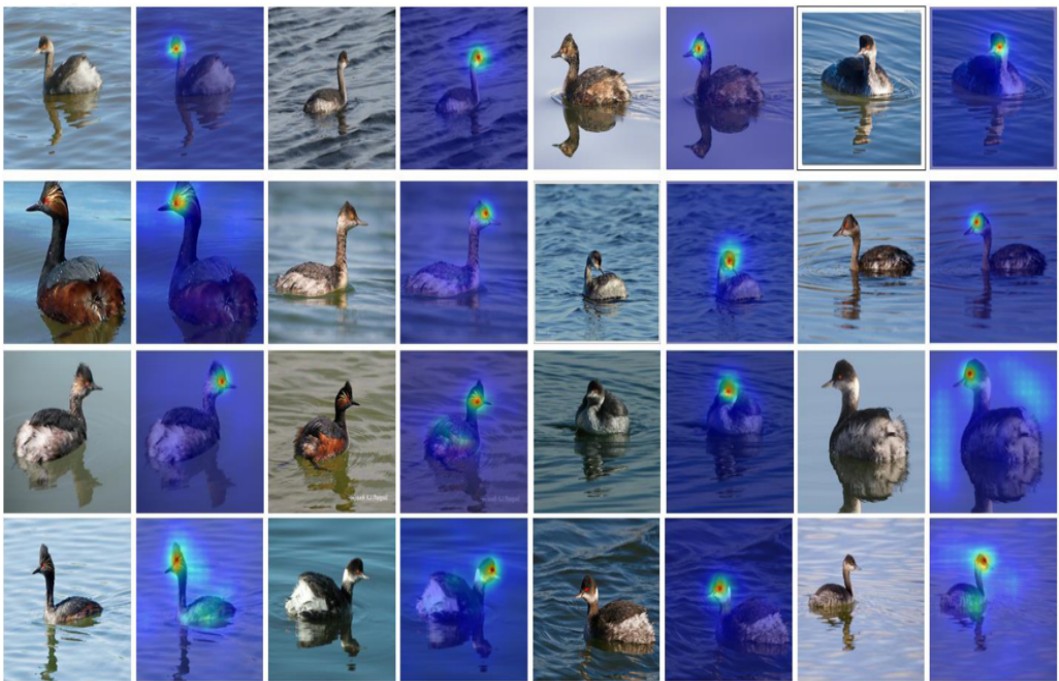

Figure 16: Eared grebe. The uniform characteristic is red eyes.

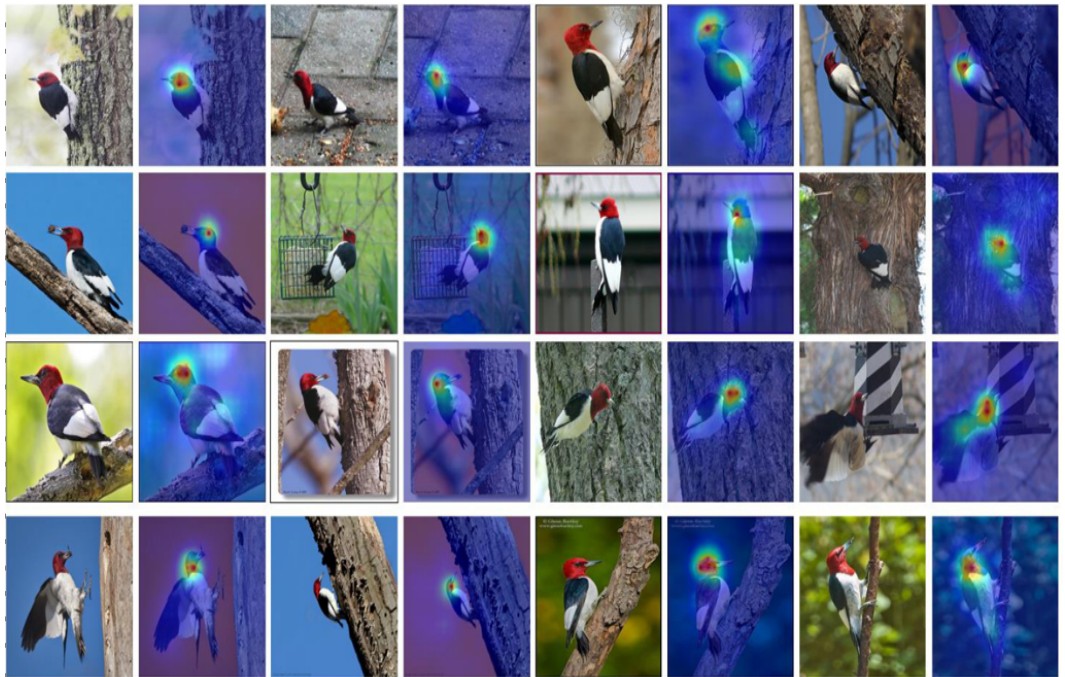

Figure 17: Red headed woodpecker. The uniform characteristic is red head.

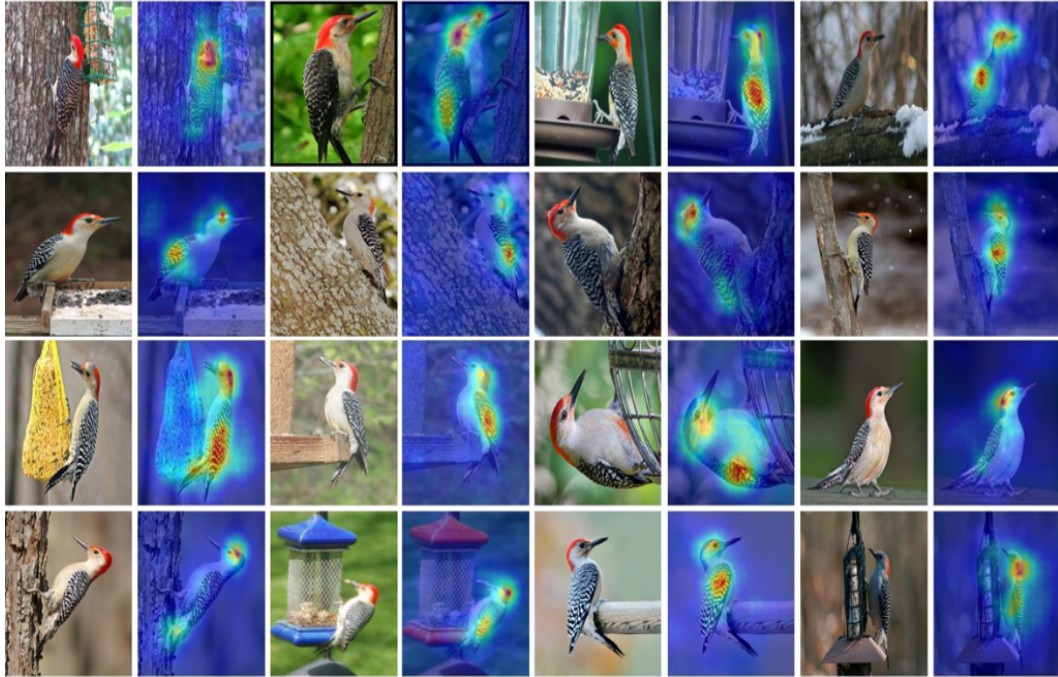

Figure 18: Red bellied woodpecker. The uniform characteristic is black and white barred patterns on their back, wings and tail. Adult males have a red cap going from the bill to the nape; females have a red patch on the nape and another above the bill.

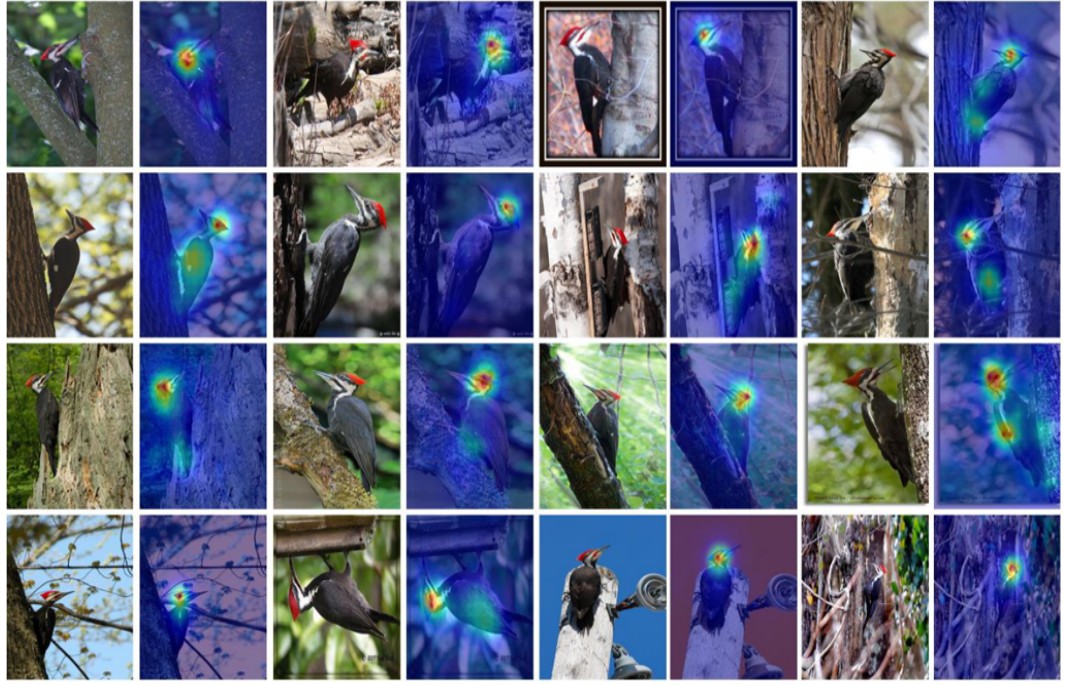

Figure 19: Pileated woodpecker. The uniform characteristic is red crest.

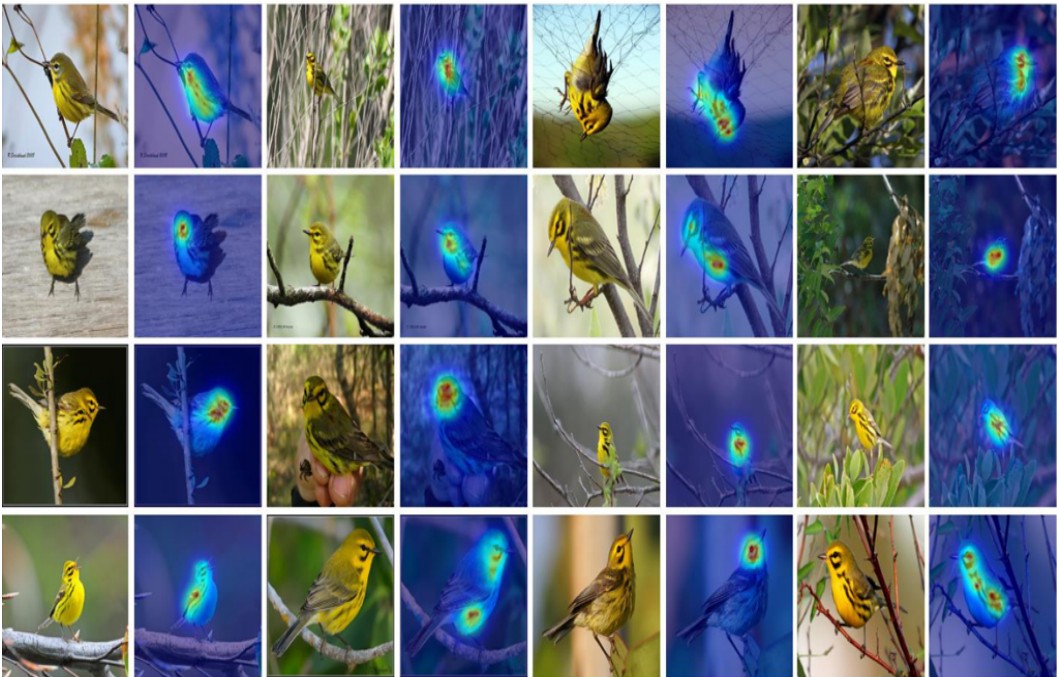

Figure 20: Prairie warbler. The uniform characteristic is dark streaks on the flanks and head.

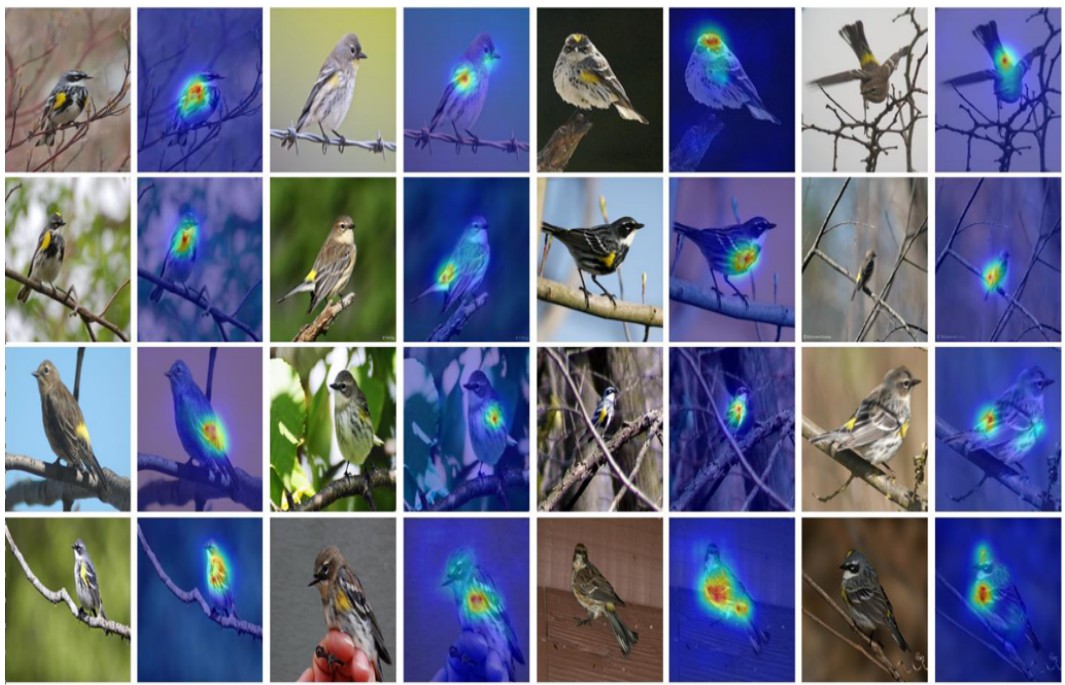

Figure 21: Myrtle warbler. The uniform characteristics are yellow crown, rump and flank patches.

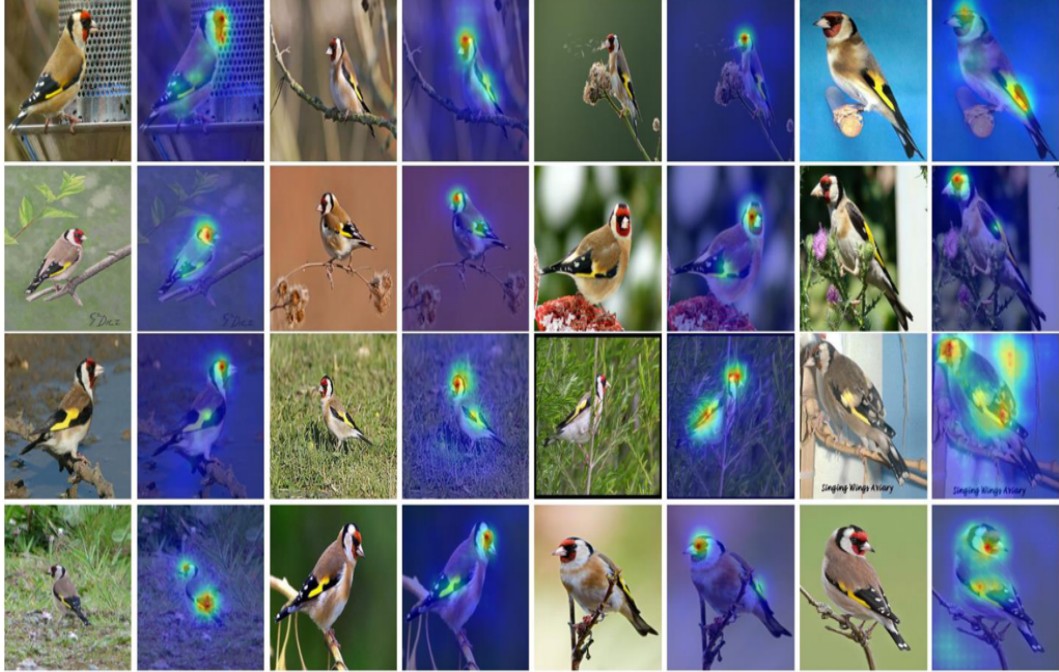

Figure 22: European goldfinch. The uniform characteristics are red face and broad yellow bar on the wings.

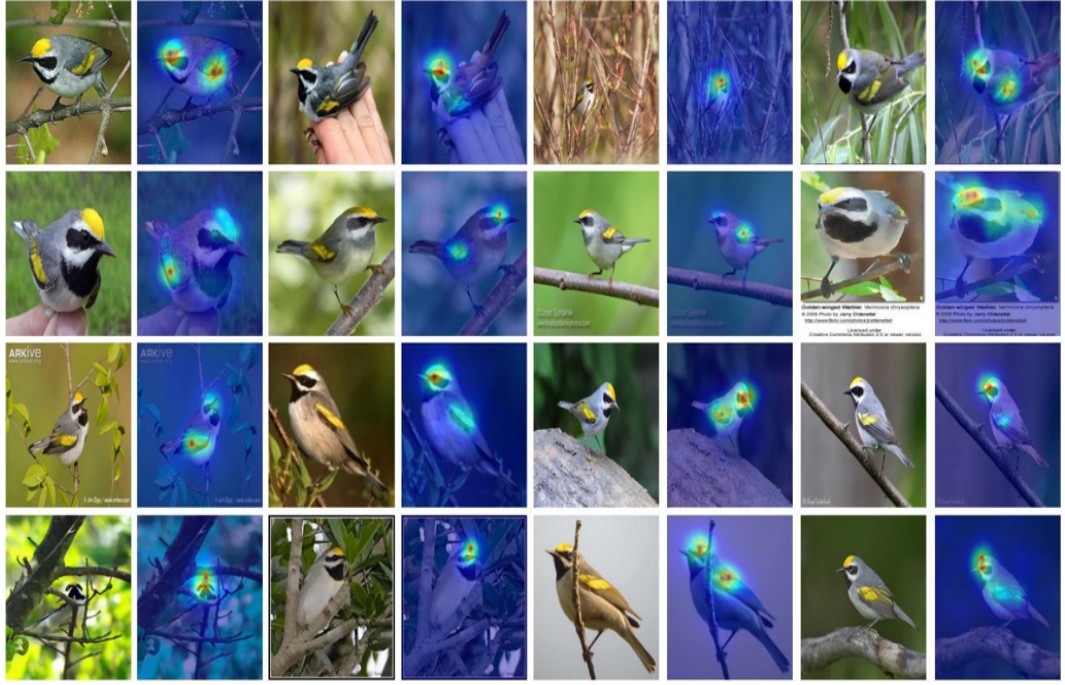

Figure 23: Gold Winged Warbler. The uniform characteristics are yellow crown and wing patch.

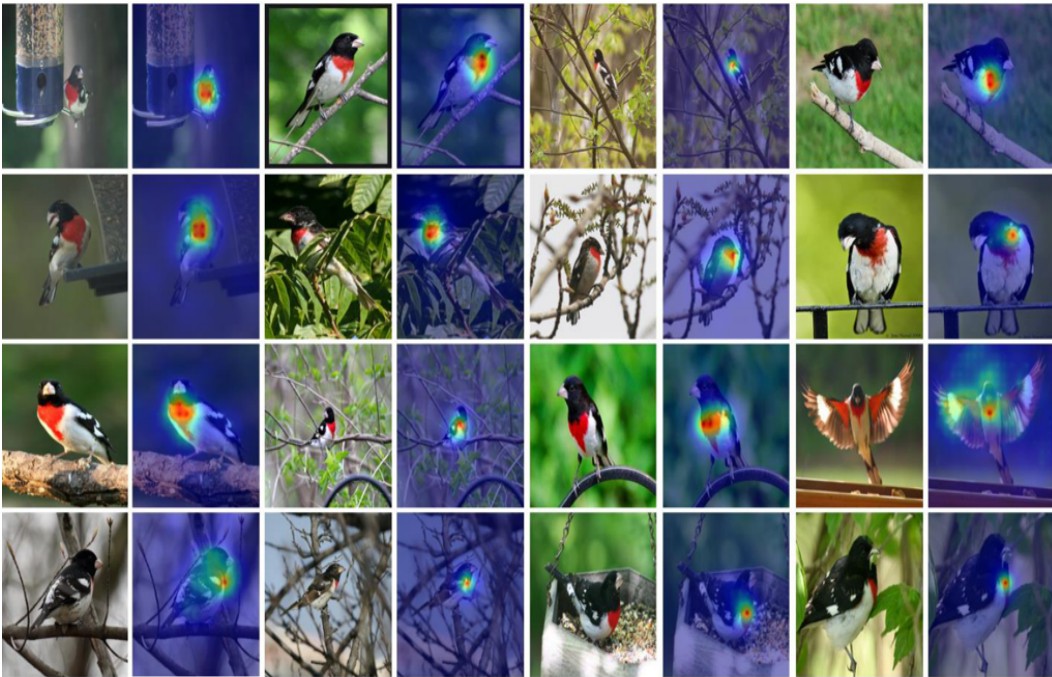

Figure 24: Rose breasted grosbeak. The uniform characteristic is bright red-red patch on the breast.

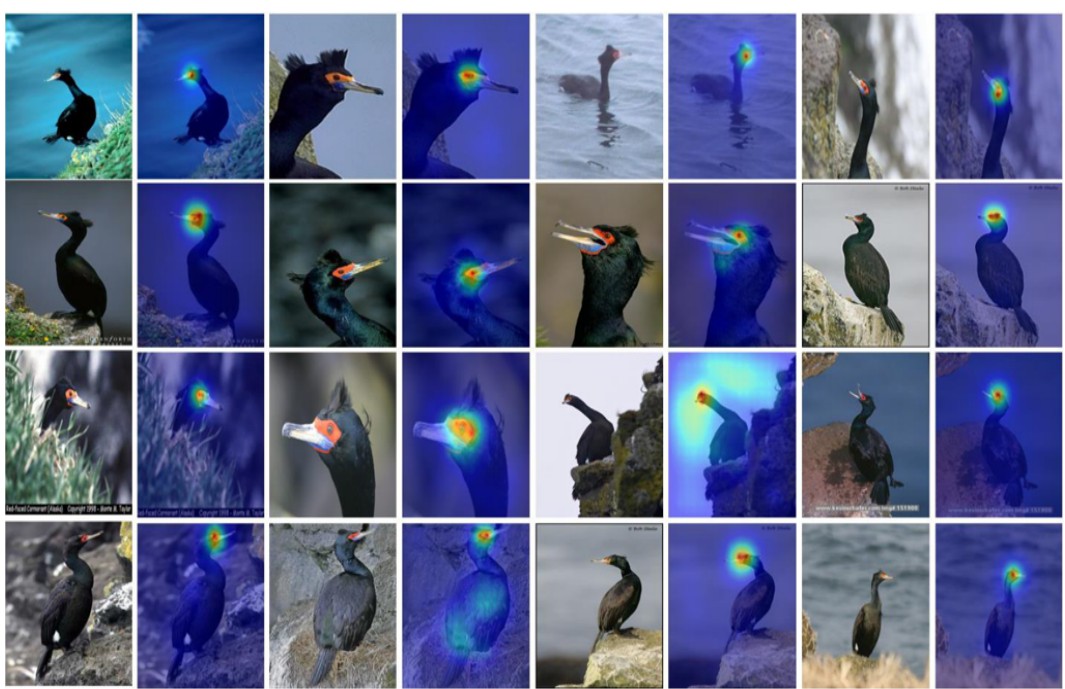

Figure 25: Red faced cormorant. The uniform characteristic is red patch around the eyes.

