# OpenReview forum: "Salient Explanation for Fine-grained Classification"
_ICLR.cc/2020/Conference — Reject_

### Official Review · AnonReviewer2 · 2019-10-18
**Official Blind Review #2**

**Rating:** 1

**Review:**

Summary:
This paper proposes a simple procedure to display the salient areas that determine classification decisions from deep networks. In practice, these area are computed by shadowing the image using a Gaussian and measuring the network contribution at every location of the image. Results in terms of "Pointing game" metric (that measure similarity between computed saliency result and ground truth) are provided on the Imagenet validation dataset.

1 Positive aspects:
- Experiments are conducted on many datasets and show visually convincing results.
- state-of-the-art on ImageNet compared to the RICE approach.

2 Negative aspects:
- Very simple approach
- Depends on many parameters (t1, t2, sigma_s, sigmal_l,  sigma_spatial)
- Improvement of PT w.r.t. Rice is not very high: 0.912 compared to 0.907.The variants Light, shadow, fusion approaches are not described in the text.
- The paper is badly written.
-- Many sentences are uninformative (e.g. abstract :  "We prove the effectiveness of GLAS for fine-grained classification using the fine-grained classification dataset")
-- There is copy-pasted text everywhere (e.g. fea - tures , formu - lated, etc.), showing there has not been proof-reading of the paper, which is pretty disconcerting to reviewers.
- There are multiple typos including misspelled names in the bib.
- blurry figures.

My initial rating is mainly motivated by the minor experimental improvement wrt RICE that is not clearly explained. There would also be a lot of work for improving the writing of the paper.

Minor : lambda is 10^-5, not 10^5 ?
is sigma_s used in the algo or any equation? Where is it defined?

**Experience Assessment:**

I have read many papers in this area.

**Review Assessment: Checking Correctness Of Derivations And Theory:**

I carefully checked the derivations and theory.

**Review Assessment: Checking Correctness Of Experiments:**

I assessed the sensibility of the experiments.

**Review Assessment: Thoroughness In Paper Reading:**

I read the paper thoroughly.

---

### Official Review · AnonReviewer3 · 2019-10-23
**Official Blind Review #3**

**Rating:** 1

**Review:**

The paper introduces a new method in the family of local perturbation-based interpretations for deep networks and more specifically for fine-grained classification tasks. Compared to other saliency map methods (gradient-based, propagation-based, etc), this family has the advantage of needing only black-box access to the model. The introduced method GLAS, scans over an image and lights/shadows each part of the image to assign an importance score to different regions of the image based on the change in model's prediction. The motivation of this work is to increase the inherently low speed of (some) methods in this family and to give better explanations by



I vote for rejecting this paper as the contributions to what already exists in the literature are not clear and the provided experimental results are not convincing.

Compared to previous perturbation-based methods, the main advantage seems to be speed. There are perturbation-based methods that do not suffer from low speed e.g. Dabkowski & Gal and give real-time perturbation-based saliency maps. The authors do not mention this work (and similar works) and do not compare both their speed and their performance against it.

One important problem (probably the most important) with the perturbation-based saliency maps is the fact that the perturbations might push a given image out of the true data manifold and therefore give an invalid interpretation of the model. The authors do not discuss the matter and how their method would address this issue. Intuitively, the introduced algorithm, more specifically the RGLAS algorithm, seems to suffer from this issue not any less than other existing methods.

The experimental results seek to demonstrate the superiority of the introduced method over rival methods. The justification behind the provided visual examples is their focus on more discriminative features (in human eyes). This does not necessarily mean that a given saliency map is more "truceful"; i.e. having a more visually appealing saliency map has nothing to do with a more truthful explanation of a model's decision making. The results focused on the mistakes of the model seem more convincing and interesting.

 The objective results first focus on the target localization metric which has traditionally been used in the literature. Although it is much faster to execute, the introduced method is only marginally superior to other methods. The most important problem, however, is that as mentioned above, there are fast methods in the literature and therefore a fair objective comparison is should include other methods as well. Secondly, the IOU measure is used. GLAS is not compared to other methods in this metric.

The authors mention the effectiveness of their work for "fine-grained" classification tasks while throughout the paper there is no convincing evidence or discussion that the method is curated for such tasks. As mentioned above, changing the scale parameter for getting more visuall appealing saliency maps is not enough evidence.

All in all, the true contribution of this work to other existing methods in this family is not enough for this venue.


A few questions and suggestions:
* How should one adjust the scale parameter? In other words, what is the hyper-parameter search scheme for this method which would make it robust against the human-biased choice of hyper-parameters which would result in visually more appealing saliency maps but not necessarily explain the model?
* Explanation of RGLAS is not clear.
* For a general reader, metrics such as IOU should be explained more clearly.
* The paper has many many typing errors.

**Experience Assessment:**

I have published in this field for several years.

**Review Assessment: Checking Correctness Of Derivations And Theory:**

I carefully checked the derivations and theory.

**Review Assessment: Checking Correctness Of Experiments:**

I carefully checked the experiments.

**Review Assessment: Thoroughness In Paper Reading:**

I read the paper thoroughly.

---

### Official Review · AnonReviewer1 · 2019-10-26
**Official Blind Review #1**

**Rating:** 3

**Review:**

This paper introduces a simple and effective method for pinpointing salient features contributing to discriminating different classes in classification. It is based on masking images using Gaussian Gaussian light and shadow (GLAS) and estimating its impact on output. The authors also develop an iterative method to identify multiple instances. Experiments quantitatively evaluate the proposed method on pointing game using ILSVRC validation set, where the proposed method outperforms recent related methods.

The technical novelty of this paper is marginal and the experimental evaluation is not convincing.

1) Marginal novelty compared to related work
There have been several masking-based black-box methods. Considering the method of RISE (Petsiuk et al., BMVC19) in particular, the novelty of the proposed method is marginal; the proposed method uses Gaussian light and shadow at individual positions while RISE uses random masks.

2) Hyperparameter sensitivity
The proposed method includes several, at least four, hyperparameters to be tuned: T, sigma_l, sigma_ss, sigma_spatial. These may affect the results significantly, but the experimental section does not discuss their sensitivity. Are they robust across different datasets? It is not clear from the experiments.

3) Experimental comparison
The comparisons are not convincing. The results on fine-grained classification are presented qualitatively, and the quantitative comparison is done only on a single dataset, ILSVRC validation set, which is very limited considering experiments in the related work, e.g., three benchmark experiments of RISE RISE (Petsiuk et al., BMVC19) on PASCAL VOC, MSCOCO, and ImageNet. Furthermore, in the ILSVRC experiments, the gap from Grad-CAM and RISE is not significant in terms of accuracy. Considering several hyperparameters of the proposed method to be tuned, these results appear less appealing. I hope the authors provide more convincing experiments, e.g., on the same benchmarks of RISE.

**Experience Assessment:**

I have published one or two papers in this area.

**Review Assessment: Checking Correctness Of Derivations And Theory:**

I carefully checked the derivations and theory.

**Review Assessment: Checking Correctness Of Experiments:**

I carefully checked the experiments.

**Review Assessment: Thoroughness In Paper Reading:**

I read the paper at least twice and used my best judgement in assessing the paper.

---

### Decision · Program_Chairs · 2019-12-19

**Decision:**

Reject

**Comment:**

This paper is interested in finding salient areas in a deep learning image classification setting. The introduced method relies on masking images using Gaussian Gaussian light and shadow (GLAS) and estimating its impact on output.

As noted by all reviewers, the paper is too weak for publication in its current form:
- Novelty is very low.
- Experimental section not convincing enough, in particular some metrics are missing.
- The writing should be improved.